# Gene-expression-based T-Cell-to-Stroma Enrichment (TSE) score predicts response to immune checkpoint inhibitors in urothelial cancer

Maud Rijnders[1,16], J. Alberto Nakauma-González [1,2,3,16],
Debbie G. J. Robbrecht [1], Alberto Gil-Jimenez [4,5], Hayri E. Balcioglu [1],
Astrid A. M. Oostvogels[1], Maureen J. B. Aarts[6], Joost L. Boormans [2],
Paul Hamberg[7], Michiel S. van der Heijden [4,8], Bernadett E. Szabados[9],
Geert J. L. H. van Leenders [10], Niven Mehra [11], Jens Voortman [12],
Hans M. Westgeest[13], Ronald de Wit[1], Astrid A. M. van der Veldt[1,14],
Reno Debets [1,17] ✉ & Martijn P. Lolkema [1,15,17]

Immune checkpoint inhibitors (ICI) improve overall survival in patients with metastatic urothelial cancer (mUC), but therapeutic success at the individual patient level varies significantly. Here we identify predictive markers of response, based on whole-genome DNA (n = 70) and RNA-sequencing (n = 41) of fresh metastatic biopsy samples, collected prior to treatment with pembrolizumab. We find that PD-L1 combined positivity score does not, whereas tumor mutational burden and APOBEC mutagenesis modestly predict response. In contrast, T cell-to-stroma enrichment (TSE) score, computed from gene expression signature data to capture the relative abundance of T cells and stromal cells, predicts response to immunotherapy with high accuracy. Patients with a positive and negative TSE score show progression free survival rates at 6 months of 67 and 0%, respectively. The abundance of T cells and stromal cells, as reflected by the TSE score is confirmed by immunofluorescence in tumor tissue, and its good performance in two independent ICI-treated cohorts of patients with mUC (IMvigor210) and muscle-invasive UC (ABACUS) validate the predictive power of the TSE score. In conclusion, the TSE score represents a clinically applicable metric that potentially supports the prospective selection of patients with mUC for ICI treatment.

Immune checkpoint inhibitors (ICIs) directed against programmed cell death protein (PD-1) or its ligand (PD-L1) have significantly improved clinical outcomes of patients with metastatic urothelial cancer (mUC). In patients with mUC with progressive disease after platinum-based chemotherapy, treatment with pembrolizumab (anti-PD-1) showed superior survival outcomes as compared to second-line chemotherapy in a phase 3 trial[1,2]. A small subset of these patients had a durable response for >2 years[3]. Furthermore, first-line treatment with pembrolizumab and atezolizumab (anti-PD-L1) showed efficacy in single-arm trials[4,5]. In addition, several clinical trials are currently investigating the efficacy of ICIs for patients with muscle-invasive bladder cancer (MIBC)[6]. Notably, the overall response rate is still limited in patients

with mUC with the accompanying risk of exposing non-responding patients to potential (severe) toxicities and expensive therapies.

To date, the only biomarker available to select patients with mUC for ICIs is PD-L1 protein in tumor tissue. However, the predictive value of PD-L1 expression heavily depends on the population of patients studied[1,4,5,7–9]. Furthermore, an important limitation of PD-L1 protein is its dependence on a specific staining platform and use of archival tumor tissue[10,11].

Another biomarker that is associated with response to ICIs is tumor mutational burden (TMB)[12,13]. Recently, high TMB (≥10 mutations per mega base-pair) was approved by the U.S. Food and Drug Administration as a pan-cancer measure to select patients with previously treated advanced solid tumors for treatment with pembrolizumab[14,15]. Furthermore, immune cell infiltration[16–18], expression of immune genes such as *IFNG, CXCL9 and CXCL10*[16,19], TGF-β signaling[20], composition of the tumor microenvironment[21], alterations in DNA damage repair (DDR) genes[22], abundance of circulating tumor DNA[23,24] and the diversity of the T cell receptor (TCR) repertoire[16,25,26] have all been associated with response and resistance to ICIs. Other studies suggest that the combination of multiple biomarkers improves response prediction for patients with mUC when compared to single biomarkers[27,28]. Collectively, there is still a general lack of evidence and validation of above-mentioned biomarkers in patients with mUC.

Along this line, we perform whole-genome DNA-sequence (WGS) and RNA-sequence (RNA-seq) data analysis and apply an integrative approach toward the discovery of new predictors for response to ICIs in patients with mUC. We identify the T cell-to-stroma enrichment (TSE) score, a transcriptomic measure comparing the expression scores of signatures for T cells and stromal resident cells and their products as a marker to predict response to anti-PD-1 in mUC. Immunofluorescence stainings, and two independent cohorts of patients with primary and metastatic UC treated with anti-PD-L1 validate the TSE score as a robust and easy-to-implement single metric that may aid to predict response to ICI monotherapy in patients with UC.

## Results

### Patient cohort and clinical characteristics

Between March 1st 2013 and March 31st 2020, 288 patients with advanced or mUC were included according to the Center for Personalized Cancer Treatment (CPCT-02) biopsy protocol (NCT01855477; Fig. 1). Fresh-frozen metastatic tumor biopsies and matched normal blood samples were collected for WGS and RNA-seq in a standardized manner[29]. Seventy patients received pembrolizumab monotherapy and were included in this analysis. Matched RNA-seq was available for 41 patients. PD-L1 combined positivity score (CPS) was assessed in biopsies of 40 patients.

One-third (*n* = 24) of patients who received pembrolizumab were responders according to response evaluation criteria in solid tumors (RECIST) v1.1. The PD-L1 CPS was positive (≥10) in 21% of responders and 24% of non-responders. Most patients (90%) received

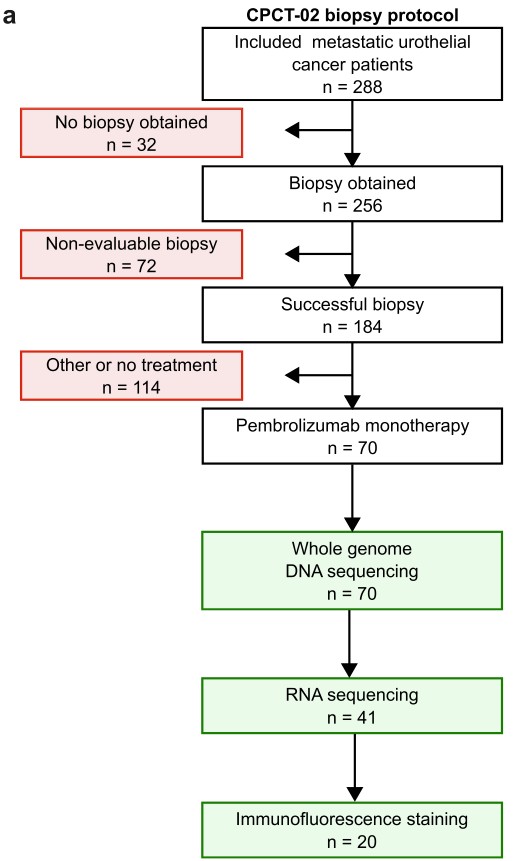

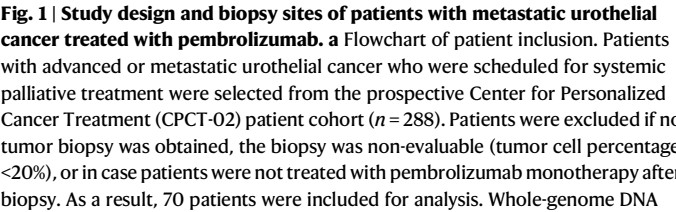

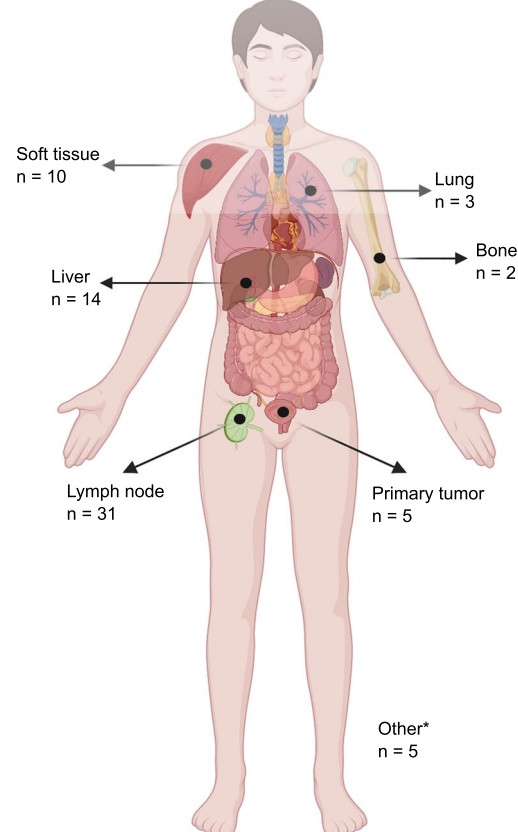

**Fig. 1 | Study design and biopsy sites of patients with metastatic urothelial cancer treated with pembrolizumab. a** Flowchart of patient inclusion. Patients with advanced or metastatic urothelial cancer who were scheduled for systemic palliative treatment were selected from the prospective Center for Personalized Cancer Treatment (CPCT-02) patient cohort (*n* = 288). Patients were excluded if no tumor biopsy was obtained, the biopsy was non-evaluable (tumor cell percentage <20%), or in case patients were not treated with pembrolizumab monotherapy after biopsy. As a result, 70 patients were included for analysis. Whole-genome DNA

sequencing (WGS) data were available for all 70 patients. Matched RNA-sequencing data were available for 41 of these patients, and tissues for immunofluorescence stainings were available for 20 of these patients. **b** Overview of the number of biopsies per metastatic site included in this study. Primary tumor samples were obtained from patients with locally advanced disease with synchronous distant metastases that were not safely accessible for a biopsy. *Other biopsy sites include adrenal gland (*n* = 2), peritoneum (*n* = 2), and local recurrence of the primary tumor (*n* = 1). Created with BioRender.com.

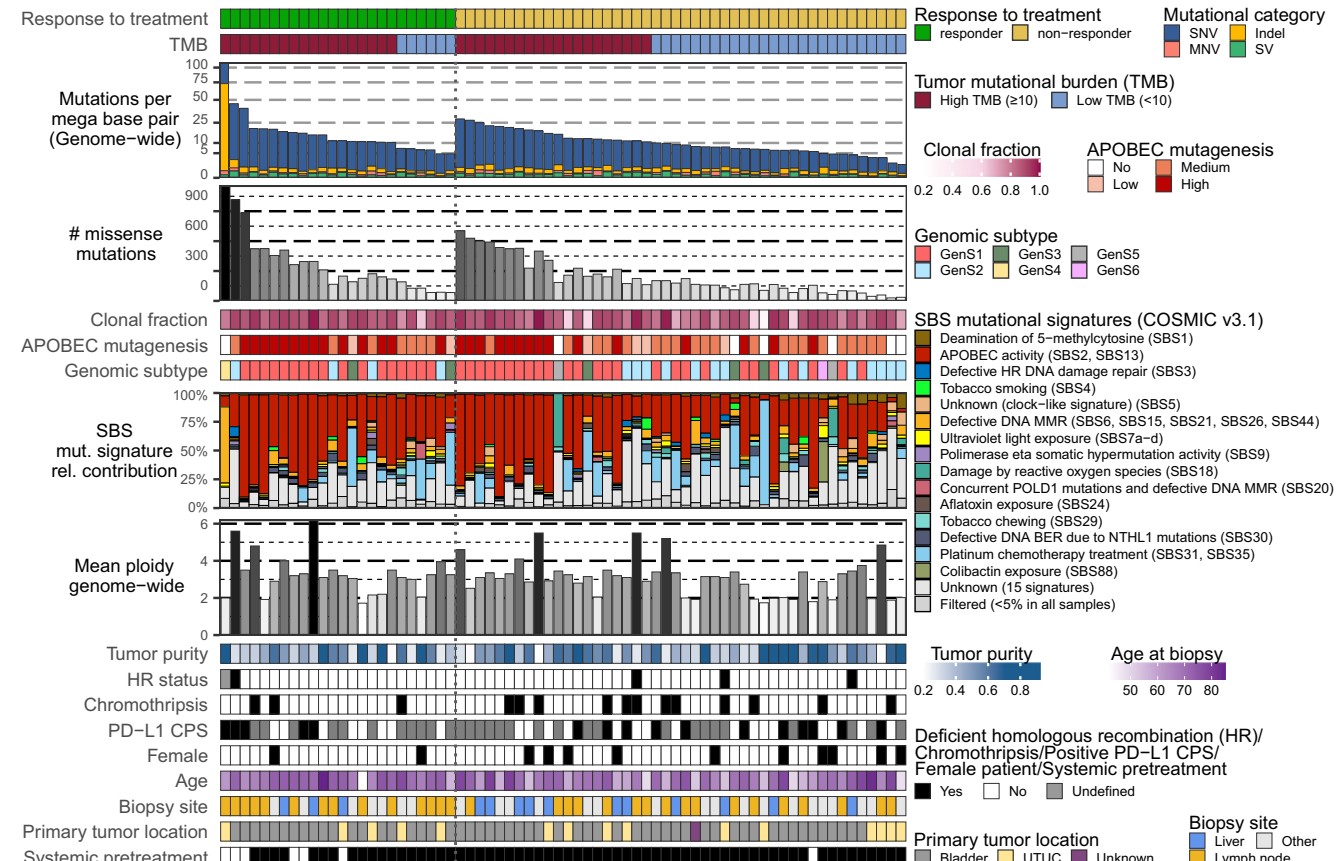

**Fig. 2 | The genomic landscape of patients with metastatic urothelial carcinoma treated with pembrolizumab.** Whole-genome sequencing data from biopsy samples of patients with metastatic urothelial carcinoma (*n* = 70) are displayed according to treatment response at 6 months of therapy (responder: ongoing complete or partial response, or stable disease, *n* = 24; non-responder: progressive disease, *n* = 46). Genomic and clinical features are listed from top to bottom as follows: genome-wide tumor mutational burden (TMB), and classification into high and low; total number of missense mutations; clonal fraction of mutations; APOBEC enrichment analysis showing tumors with no-, low-, medium- and high-APOBEC mutagenesis; genomic subtypes according to mutational signatures[30]; single base substitution (SBS); mutational signatures according to COSMIC v3.1; genome-wide mean ploidy; tumor purity; homologous recombination (HR) status; tumors with at least one chromothripsis event; PD-L1 combined positivity score (CPS) according to the companion diagnostic assay of pembrolizumab (positive: CPS ≥ 10, negative: CPS < 10, or not available (NA)); female patients; age at time of biopsy; metastatic site from which a biopsy was obtained; primary tumor location (bladder or upper tract urothelial carcinoma, UTUC); and patients who received systemic treatment prior to start of anti-PD-1 therapy. Source data are provided as a Source Data file.

pembrolizumab as second-line therapy, but responders more frequently received pembrolizumab as first-line therapy compared to non-responders (25% vs. 2%; two-sided Fisher's exact test *p* = 0.005; chemotherapy-naïve patients were selected for a positive PD-L1 CPS). At data cut-off, 27% of patients were alive. The median overall survival (OS) was 8.9 months, and the median progression-free survival (PFS) was 2.9 months. Patient characteristics are summarized in Supplementary Table 1.

### TMB and APOBEC mutagenesis only modestly predict response to pembrolizumab

The majority of patients (54%) in our cohort had a high TMB (Fig. 2). Of patients with high TMB, 47% were responders, whereas only 19% of patients with low TMB were responders (two-sided Fisher's exact test *p* = 0.022; Supplementary Fig. 1). Previously, five genomic subtypes (GenS) of mUC have been identified according to COSMIC v3.1 mutational signatures[30]. GenS1, which is related to APOBEC mutagenesis, was identified in 61% of samples. Overall, genomic subtypes were not associated with treatment response. Of patients with high APOBEC mutagenesis (*n* = 29), 48% responded to pembrolizumab, whereas 24% of patients with non-high APOBEC mutagenesis (*n* = 41) responded to pembrolizumab (two-sided Fisher's exact test *p* = 0.045; Supplementary Fig. 1). One responder had no evidence of APOBEC mutagenesis

but had a high TMB as a result of defective DNA mismatch repair. We did not observe differences between responders and non-responders with respect to HR deficiency nor presence of chromothripsis.

Furthermore, when evaluating the presence of driver gene alterations, we did not observe statistically significant differences between responders and non-responders (Supplementary Fig. 2). Alterations in canonical signaling pathways were most frequently observed in the p53, cell cycle, and RTK-RAS pathways (Supplementary Fig. 3a and Supplementary Data 1), yet not significantly different between the two patient groups. Also, the frequency of alterations in DDR genes and signaling pathways was not statistically different between responders and non-responders (Supplementary Fig. 3b). Activity of the p53 pathway was reduced in those patients (responders and non-responders alike) with genomic alterations in this pathway (Supplementary Fig. 3c). Collectively, the genomic analyses revealed only modest predictive value of TMB and APOBEC mutagenesis for response to anti-PD-1.

### Expression of genes representing immune cells and stromal cells distinguishes responders from non-responders to pembrolizumab

Differential gene expression analysis of RNA-seq data (*n* = 41) revealed that up-regulated genes in responders vs. non-responders were part of

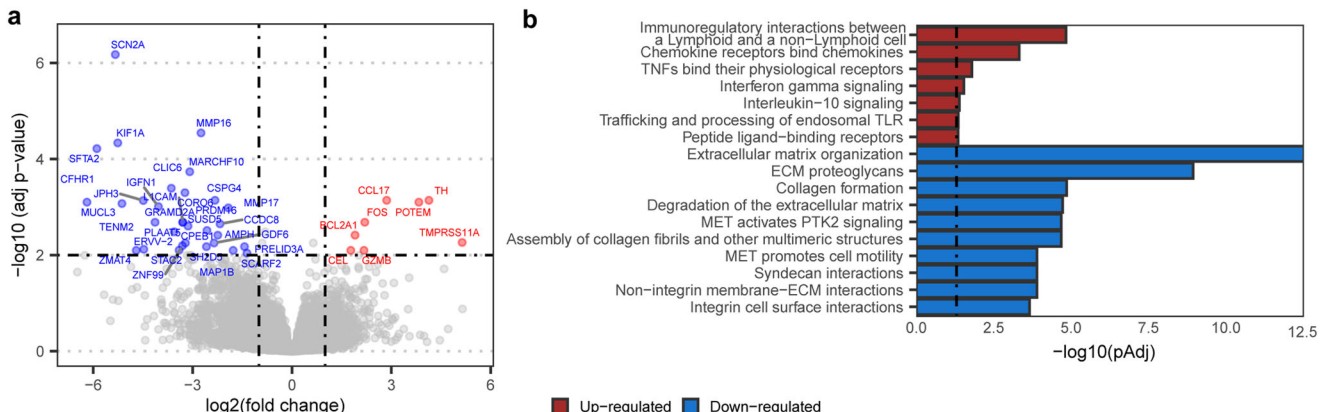

**Fig. 3 | Differential expression of genes and pathways related to immune cell and stromal cell activity for responders and non-responders to pembrolizumab. a** Volcano plot showing genes with up-regulated or down-regulated expression in responders ($n = 13$) vs. non-responders ($n = 28$). Genes of which differential expression analysis showed adjusted $p < 0.01$ and absolute log2 fold change >1 are labeled in red (up-regulated) and blue (down-regulated). Differential expression analysis of transcripts was performed using the Wald test. **b** Bar diagrams specify the pathways of differentially expressed genes according to the hypergeometric distribution calculated with ReactomePA v1.34.0[42]. Enriched pathways with adjusted $p < 0.05$, indicated by the vertical dashed line, were considered significant. All significantly up-regulated pathways, and the top ten down-regulated pathways are displayed. In (**a**) and (**b**), $p$ values were adjusted using the Benjamini–Hochberg method. Source data are provided as a Source Data file.

a chemokine pathway, and a pathway related to interactions between lymphoid and non-lymphoid cells (Fig. 3). Down-regulated pathways in responders (up-regulated in non-responders) were related to extracellular matrix organization and collagen formation, generally linked to the activity of stromal cells. An example of an up-regulated pathway in responders involved interleukin-10 (IL-10), a recognized immunosuppressor, which in recent studies has also been associated with T cell activation in solid tumors[31]. Since IL-10 can be expressed by several cell types, including cancer cells, the origin as well as the exact functioning of IL-10 in the context of ICI treatment requires further investigation.

## Patient stratification according to T cell-to-stroma enrichment score coincides with response to pembrolizumab

Following up on the pathway analysis displayed in Fig. 3, we have interrogated the transcriptomic landscape of our cohort for a broad list of gene signatures related to T cells, other (non-T cell) immune cells, and stromal cells and their products (see Supplementary Table 2 for a detailed overview of gene signatures). Some of these signatures have been reported as predictors of response and resistance to ICIs[18,20]. Hierarchical clustering according to the complete set of signatures revealed three distinct patient clusters (Fig. 4). In cluster one ($n = 18$), 61% of patients showed a response to pembrolizumab. These patients predominantly had high signature scores for T cells and other immune cells. In cluster two ($n = 10$), 20% of patients showed a response to pembrolizumab. These patients generally had a similar score for all signatures, independent of the cell type(s) and products they represented. In cluster three ($n = 13$), none of the patients showed a response to pembrolizumab. These patients predominantly had high signature scores for stromal cells and their products. The above clustering suggested that signature scores for immune cells and stromal cells and their products were related to response to pembrolizumab. To select those signatures with the most predictive value, ROC curves were constructed per signature, which demonstrated areas under the curve (AUC) that ranged from 0.54 to 0.77 (median = 0.68; Supplementary Table 3). The highest AUCs (>0.7) were observed for T cells and stromal cells and their products, and (non-T cell) immune cells showed AUCs below the median. Signatures that showed the highest discriminatory power were selected and combined into either a global T cell or a global stromal signature (Supplementary Fig. 4). Notably, logistic regression analyses showed that the global T cell signature was an independent predictor of response (Coefficient = 3.03, $p = 0.005$), while the global stromal signature was an independent predictor of non-response (Coefficient = −2.40, $p = 0.010$) to pembrolizumab. Next, we combined these two global signatures into a single metric that we termed the T cell-to-stroma enrichment (TSE) score and that reflects the abundance of T cells relative to that of stromal cells and their products. This TSE score revealed a significantly higher predictive value (AUC = 0.88) for treatment response than either global or individual signatures alone (Supplementary Table 3). Stratifying patients by their TSE score resembled the patient groups obtained by hierarchical clustering and revealed almost identical response rates (67, 21 and 0% for patients with a positive, neutral or negative TSE score).

It is noteworthy that patients with a positive TSE score were enriched for biopsies from lymph nodes (Fisher's exact test $p < 0.001$). When the analysis was restricted to only samples from lymph nodes ($n = 18$), the predictive value of the TSE score reached similar statistics as for the whole cohort (two-sided Fisher's exact test $p = 0.02$; Supplementary Fig. 5), demonstrating the robustness of the TSE score. Patients with a neutral TSE score were enriched for females (two-sided Fisher's exact test $p = 0.004$), whereas other characteristics such as age (Kruskal–Wallis test by ranks, $p = 0.64$) and pre-treatment status (two-sided Fisher's exact test $p = 0.54$) did not correlate with the TSE score categories. The vast majority of tumors with a negative TSE score (92%) were classified as stroma-rich or basal/squamous according to the transcriptomic subtypes of mUC[30]. TMB and APOBEC mutagenesis were not different between the three TSE score groups (Fig. 4). Likewise, the distribution of driver gene alterations, hotspot mutations and gene fusions were similar across TSE score groups (Supplementary Fig. 6). Also, PD-L1 CPS was similar across the TSE score groups (Fig. 4), whereas *CD274* (PD-L1) and *PDCD1* (PD-1) gene expressions were higher for patients with a positive vs. negative TSE score (Supplementary Fig. 7). When assessing the relative abundance of immune cell populations, we observed that the fraction of myeloid dendritic cells was higher in patients with a positive vs. negative TSE score (Supplementary Figs. 8 and 9). Furthermore, the TCR diversity and the relative abundance of less frequent TCR clonotypes was higher in patients with a positive vs. negative TSE score (Fig. 4 and Supplementary Figs. 8 and 10).

## The TSE score is a superior predictor for response and survival compared to genomic metrics

To evaluate the predictive values of the TSE score, TMB, APOBEC mutagenesis and their combinations for response to pembrolizumab,

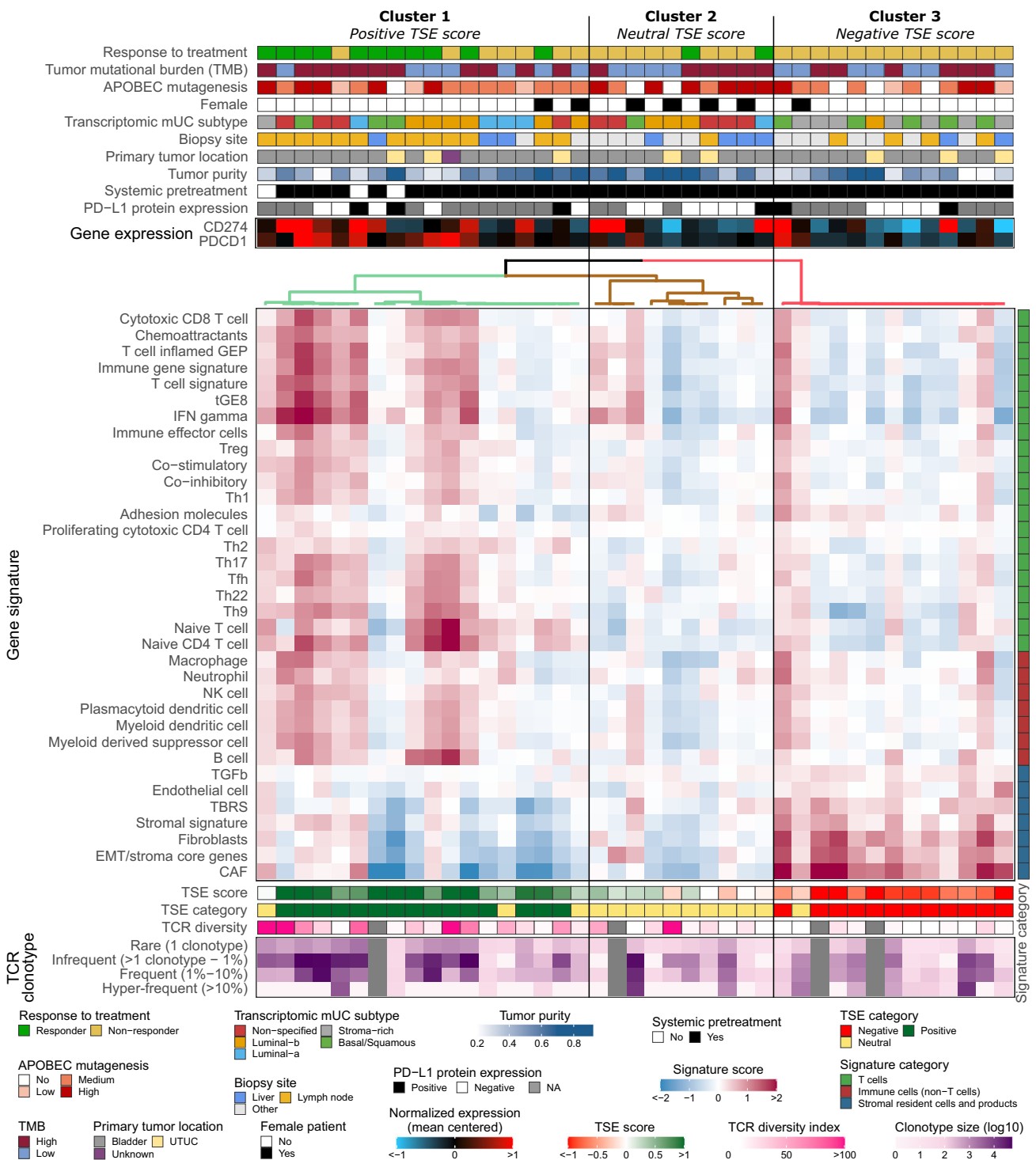

**Fig. 4 | Hierarchical clustering of gene signatures representing T cells, immune cells and stromal cells and their products distinguishes responders from non-responders to pembrolizumab.** Transcriptomic profile of 41 patients with metastatic urothelial carcinoma (mUC), clustered using ConsensusClusterPlus v1.54.0[38] according to gene signature scores. Transcriptomic and clinical features are listed from top to bottom as follows: response to treatment at 6 months of therapy (responder: ongoing complete or partial response, or stable disease, n = 13; non-responder: progressive disease, n = 28); tumor mutational burden (TMB) classified into high and low; APOBEC enrichment analysis showing tumors with no-, low-, medium- and high-APOBEC mutagenesis; transcriptomic subtypes of mUC[30]; biopsy site; primary tumor location (bladder or upper tract urothelial carcinoma, UTUC); tumor purity; patients who received systemic treatment prior to start of anti-PD-1 therapy; PD-L1 combined positivity score (CPS; positive: CPS ≥ 10, negative: CPS < 10, or not available (NA)); *CD274* (PD-L1) and *PDCD1* (PD-1) gene expression; expression score for reported gene signatures related to T cells, immune cells (non-T cells), and stromal cells and their products; T cell-to-stroma enrichment (TSE) score; categories of the TSE score (positive, neutral or negative); T cell receptor (TCR) diversity index and clonotype sizes. Source data are provided as a Source Data file.

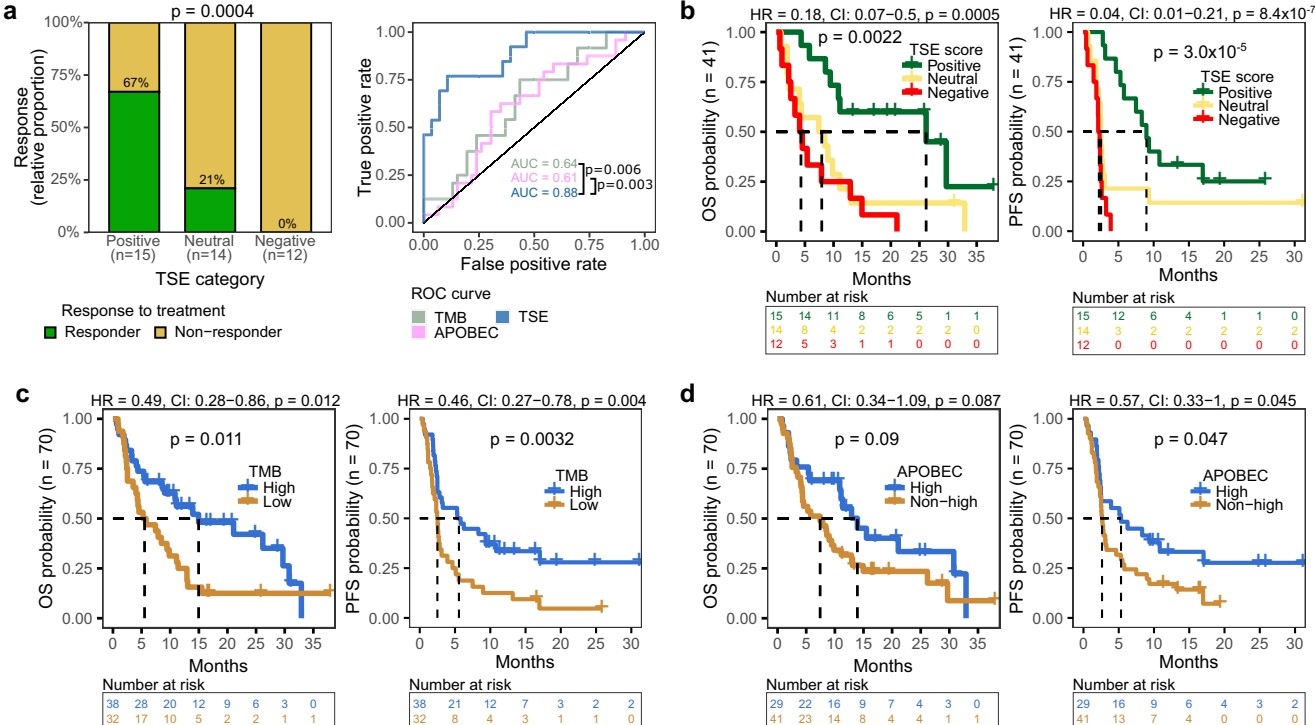

**Fig. 5 | Association of the TSE score, TMB and APOBEC mutagenesis with response to pembrolizumab and overall and progression-free survival. a** Bar graphs display the relative proportion of responders and non-responders in patients with a positive, neutral, or negative TSE score (TSE positive, $n = 15$; TSE neutral, $n = 14$; TSE negative, $n = 12$). $p$ value of TSE positive vs. negative was determined using the two-sided Fisher's exact test. Receiver operating characteristic (ROC) curves of TSE score, TMB, APOBEC mutagenesis (enrichment for APOBEC-associated mutations), and their combinations were constructed using continuous variables. The area under the curve (AUC) is displayed per condition, and $p$ values reflect DeLong's test of AUC's. **b** Overall survival (OS) and progression-free survival (PFS) probability in patients with a positive, neutral or negative TSE score; (**c**) high ($n = 38$) or low ($n = 32$) TMB; or (**d**) high ($n = 29$) or non-high ($n = 41$) APOBEC mutagenesis. Log-rank test was applied to survival curves. For TSE score, hazard ratio (HR) was calculated for positive vs. negative. CI confidence interval. Source data are provided as a Source Data file.

ROC curves were analyzed (Fig. 5a). The TSE score was superior to TMB or APOBEC mutagenesis to identify responders from non-responders (Fig. 5a; DeLong's test $p = 0.006$ and $p = 0.003$ for AUC of TSE score vs. TMB and APOBEC mutagenesis, respectively). The AUC of the TSE score did not improve when combined with TMB and/or APOBEC mutagenesis. Furthermore, patients with a positive TSE score had a longer overall survival (OS) and progression-free survival (PFS) when compared to other patients (Fig. 5b). Multivariate cox regression analysis, using continuous values, showed that the TSE score had a superior predictive value for OS (TSE score $p < 0.001$; TMB $p = 0.21$; APOBEC $p = 0.25$) and PFS (TSE score $p = 0.002$; TMB $p = 0.32$; APOBEC $p = 0.27$) than TMB and APOBEC mutagenesis (Fig. 5b–d).

In extension to the transcriptomic analysis, we evaluated the TSE score at protein level. To this end, we performed immunofluorescence stainings to visualize and quantify CD4 and CD8 T cells as well as fibroblast activating protein (FAP) and podoplanin (PDPN) as stromal products (see "Materials and methods" for details) using 20 samples with matched RNA-seq data (Fig. 6 and Supplementary Fig. 11). In tumor tissues, CD4 and CD8 T cell markers (present in TSE-positive; nearly absent in TSE-negative samples) showed an inverse relationship with PDPN expression (nearly absent in TSE-positive; present in TSE-negative samples) (Fig. 6a and Supplementary Fig. 11a, b). When calculating a protein-based metric according to the TSE-RNA score, we observed that the TSE-protein score correlates well with its RNA-based counterpart (Fig. 6b). Finally, we confirmed that combining the protein markers for T cells and stromal products into a single metric improves prediction for response to pembrolizumab (Fig. 6b and Supplementary Fig. 11c–e).

### The TSE score as a predictor for response to pembrolizumab was validated in independent cohorts of patients with urothelial cancer

To substantiate the predictive value of the TSE-RNA score for response to ICIs, and its potential clinical applicability, we set out to validate this score in two independent cohorts of UC patients from the IMvigor210[20] ($n = 348$) and ABACUS[18] ($n = 84$) trials. The IMvigor210 trial evaluated the efficacy of atezolizumab (anti-PD-L1) in patients with platinum-refractory locally advanced or mUC. The TSE score was predictive for response (based on best overall response according to RECIST v1.1) to anti-PD-L1 in this cohort. It is noteworthy that for this trial the AUC of the TSE score (AUC = 0.65) was similar to the AUC of TMB (AUC = 0.71). Patients with a positive TSE score had a higher response rate (36%) than patients with a neutral (18%) or negative (13%) TSE score. A longer OS was observed in patients with a positive TSE score when compared to other patients (Fisher's exact test $p < 0.001$) (Fig. 7a). Interestingly, among non-responders in this trial, there was an enrichment for a negative TSE score in pre-treated (30/59) vs. treatment-naïve (9/37) patients (two-sided Fisher's exact test $p = 0.01$; Supplementary Fig. 12), which implies that the micro-milieu of tumors has evolved toward relative T cell deficiency as a consequence of pre-treatment. The ABACUS trial evaluated the efficacy of neoadjuvant treatment with atezolizumab in patients with MIBC. Again, the TSE score was predictive for response (based on a pathological complete response (pCR) at cystectomy) in the second validation cohort. In the ABACUS cohort, TMB failed to predict response to neoadjuvant treatment[18], and the AUC for the TSE score (AUC = 0.74) was higher than the AUC of TMB (AUC = 0.51). The pCR rate was 44% for patients with a positive TSE score and was higher when compared to patients

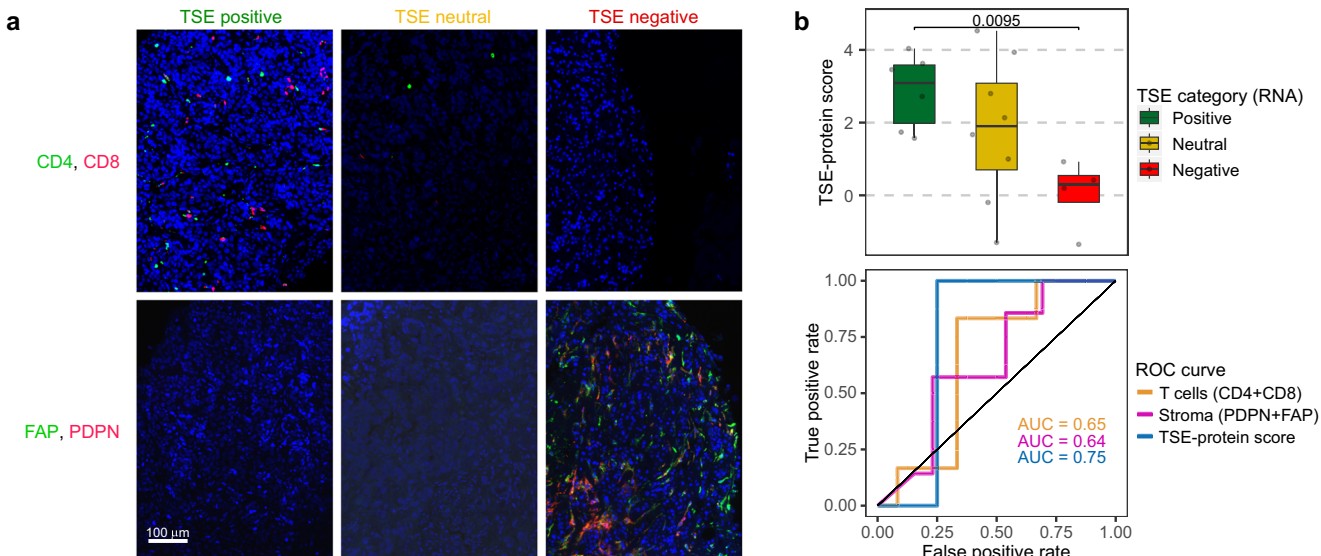

**Fig. 6 | Immunofluorescence staining of T cell and stromal markers capture the TSE score. a** Representative images of immunofluorescence staining for two T cell markers (CD4 and CD8 T cells) and two markers of stromal resident cells and their products (FAP and PDPN) according to the T cell-to-stroma enrichment (TSE) score (TSE positive, *n* = 7; TSE neutral, *n* = 8; TSE negative, *n* = 5). **b** The TSE-protein score was calculated from the densities of the four markers from (**a**) in analogy to the TSE-RNA score; see "Materials and methods" for details. Upper plot: box plot displaying the TSE-protein score in 18 patients with a positive, neutral, or negative TSE-RNA score. The TSE score for two patients is missing because the quantification of T cell markers was unsuccessful. Box plots show median, inter-quartile range (IQR: Q1–Q3) and whiskers (1.5xIQR from Q3 to the largest value within this range or 1.5xIQR from Q1 to the lowest value within this range). Two-sided Wilcoxon-rank sum test was applied to compare TSE-protein positive vs. TSE-protein negative samples. Lower plot: receiver operating characteristic (ROC) curves of T cell markers (*n* = 18), stromal markers (*n* = 20) and the TSE-protein score (*n* = 18). The area under the curve (AUC) is displayed per condition. Source data are provided as a Source Data file.

with a negative TSE score (9%, two-sided Fisher's exact test *p* = 0.009). In addition, patients with a positive TSE score experienced a longer recurrence-free survival (Fig. 7b). Together, these results suggest that contrary to TMB or ABOPEC mutagenesis, the TSE score is a robust marker that predicts response to anti-PD-1 as well as anti-PD-L1 in both metastatic and primary UC.

## Discussion

In this study, we aimed to identify a marker or metric that predicts response to pembrolizumab by analyzing the genomic and transcriptomic profiles of metastatic lesions from patients with mUC prior to treatment. We observed that gene expression signatures of T cells or stromal cells and their products associated with either response or resistance to pembrolizumab. We translated these findings into the TSE score, a single transcriptomic metric that captures individual and already recognized gene signatures related to abundance of T cells and stromal cells and their products. This TSE score acted as a superior predictor for response and survival when compared to alternative markers, and this score was not confounded by metastatic site. Furthermore, the predictive value of the TSE score was supported by immunofluorescence stainings in tumor tissue, and was validated in two independent cohorts of patients with primary and metastatic urothelial cancer treated with anti-PD-L1.

In line with previous studies in patients with mUC[13,16,25,26], high TMB and high APOBEC mutagenesis were associated with response to pembrolizumab in our cohort. However, the predictive value of both genomic scores was limited since ~20% of patients with low TMB or non-high APOBEC mutagenesis still had benefit from treatment. PD-L1 CPS failed to predict outcome in our cohort, although we cannot exclude that the analysis may have been underpowered, as other studies have shown a link between PD-L1 CPS and response to immunotherapy[11]. Analysis of transcriptomics revealed that expression of genes representing immune cells and stromal cells distinguishes responders from non-responders to pembrolizumab, particularly

those that were part of chemotaxis, interactions between lymphoid and non-lymphoid cells, and extracellular matrix organization. The TSE score, taken into account signatures that capture T cells and stromal cells and their products, resulted in a better predictive value when compared to TMB, APOBEC or single gene signatures. In fact, the majority of patients with a positive TSE score responded to pembrolizumab and patients had superior OS and PFS when compared to other patients. In contrast, none of the patients with a negative TSE score had a response to treatment. At the transcriptomic level, tumors with a negative TSE score were characterized by signatures related to TGF-β signaling and epithelial-to-mesenchymal transition (EMT), and most of these tumors were of the stroma-rich or basal-squamous mUC subtype. A negative TSE score may reflect an immune-evasive state limiting T cell influx and migration caused by an overly active stromal compartment. Indeed, TGF-β signaling has previously been associated with an immune excluded phenotype, and a fibroblast and collagen-rich tumor stroma in anti-PD-L1 resistant mUC[20]. In addition, in patients with mUC treated with anti-PD-1, EMT-like gene expression by stromal cells was shown to be related to treatment resistance, even in the presence of T cell infiltration[32]. Moreover, the association between non-response as well as poor survival and a fibrotic subtype of the tumor micro-environment has been observed in patients with mUC and other cancers[21].

Early studies have shown an association between TCR repertoire and response to ICI[33,34]. In the current study, we found that patients with a positive TSE score showed higher TCR diversity and higher abundance of infrequent TCR clonotypes, whereas patients with a negative TSE score showed higher abundance of hyper-frequent TCR clonotypes. These data support the notion that in tumor tissues with higher abundance of T cells over stromal resident cells, and consequently more contact areas between T cells and tumor cells, T cell expansion would easily occur and result in a relative dominance of infrequent TCR clones (as dictated by antigens expressed by the tumors). In contrast, in tumor tissues with higher abundance of

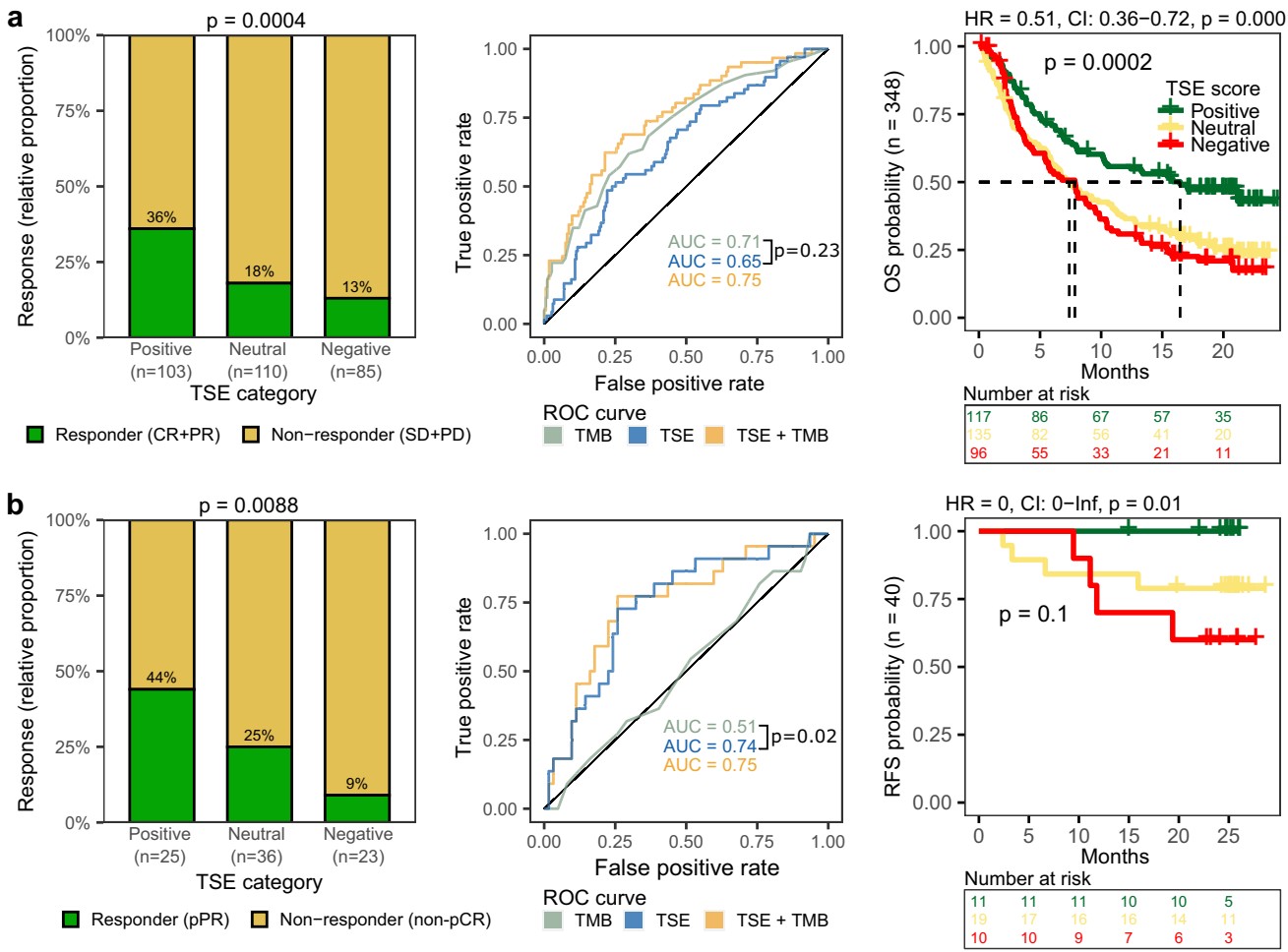

**Fig. 7 | Predictive value of the TSE score for response to ICIs in two independent cohorts of patients with urothelial carcinoma.** Validation of the T cell-to-stroma enrichment (TSE) score in the (**a**) IMvigor210 cohort (*n* = 348) and the (**b**) ABACUS trial (*n* = 84). Left graphs: the bar graphs display the relative proportion of responders and non-responders in patients with a positive, neutral, or negative TSE score. In the IMvigor210 cohort (*n* = 298 response to treatment available), responders were defined as those patients with a complete response (CR) or partial response (PR), and non-responders as those with stable disease (SD) or progressive disease (PD) as best overall response according to RECIST v1.1. In the ABACUS trial, responders were patients with a pathological complete response (pCR) at cystectomy. Two-sided Fisher's exact test was applied on the proportion of responders in patients with a positive vs. negative TSE score. Middle graphs: receiver operating characteristic (ROC) curves of the TSE score, tumor mutational burden (TMB) and their combination. *p* values reflect DeLong's test of area under the curve (AUC) generated for the TSE score vs. TMB (NS not significant). Right graphs: overall survival (OS) probability was available for all patients in the IMvigor210 cohort and recurrence-free survival (RFS) was available for 40 patients in the ABACUS cohort. Log-rank test was applied to survival curves. Hazard ratios (HR) were calculated for patients with a positive vs. negative TSE score. CI confidence interval.

stromal resident cells over T cells, T cell expansion would be restricted, which, given the low total size of TCR clones, would yield a relative dominance of hyper-frequent TCR clones. In addition, samples with a positive TSE score contained a higher fraction of dendritic cells when compared to samples with a negative TSE score. This finding extends our previous study where we reported the clustering of myeloid cells and T cells in metastatic lesions of patients who responded but not patients who did not respond to anti-PD1[35].

The TSE score was visualized and tested at the protein level by staining for T cell markers and stromal products. When calculating a protein-based TSE metric, we again observed clear predictions for response to pembrolizumab. These findings support the TSE score as a transcriptomic metric, not excluding the potential clinical application of the TSE-protein score, which would require additional studies into the most optimal combination of protein markers and their cut-offs to stratify patients according to TSE-protein categories. Furthermore, the TSE score was validated in two independent patient cohorts, namely patients with mUC treated with atezolizumab (IMvigor210 trial) and patients with MIBC treated with neo-adjuvant atezolizumab (ABACUS

trial). The TSE score was able to predict response to atezolizumab in both cohorts, and was associated with improved survival. We also utilized the IMvigor210 cohort to assess whether response prediction according to the TSE score is related to pre-treatment. We observed that non-responders who had received platinum-based chemotherapy were enriched for a negative TSE score when compared to those patients who had not received pre-treatment. This suggests that platinum-based chemotherapy induced a micro-environmental shift toward less T cells and/or more stromal resident cells, and adversely impacts response to ICI as a second-line therapy, and warrants confirmative analyses using paired samples before and after chemotherapy. The predictive value of the TSE score for OS in the IMvigor210 cohort appeared less strong compared to our cohort. Possibly this can be explained by differences between the two cohorts with respect to sample collection, definition of response to treatment, and/or timing of tumor tissue collection relative to treatment initiation. In the ABACUS cohort, and in line with the current cohort, tissue samples were obtained directly prior to therapy initiation and may therefore better reflect the transcriptomic state of the tumor, suggesting that fresh

biopsies may improve the predictive power of the TSE score. Importantly, based on the findings from the ABACUS cohort, the TSE score seems to be applicable beyond the metastatic setting, confirming the robustness of the TSE score as a predictor for response to ICIs in patients with urothelial cancer. A limitation of the current study is the relatively small cohort size, which reduced our statistical power to further improve the stratification of patients within the TSE score groups. More specifically, the group of patients with a neutral TSE score showed a response rate of ~20% in all three independent cohorts. Identifying responders within this group using genomics, transcriptomics and other molecular markers, would be necessary to improve the selection of these patients for ICIs.

Limitations of this study include the small cohort size and sample heterogeneity regarding metastatic site and systemic pre-treatment. A larger cohort is necessary to confirm our findings. In addition, predictive studies that include the TSE score together with other genetic, molecular and clinical variables may further improve the selective value of the TSE score.

In conclusion, analysis of the transcriptome and supported by immune stainings identified the TSE score as a clinically relevant marker to select patients with UC for PD-(L)1-targeting ICIs, both in the primary and metastatic setting. Since a negative TSE score identifies patients who will not derive benefit from treatment with PD-(L)1-targeting ICIs, future studies are warranted to adapt treatment for these patients in order to improve outcomes.

## Methods

### Patient cohort and study design
Between March 1st 2013 and March 31st 2020, patients with advanced or mUC from 31 Dutch hospitals were included in the nationwide Center for Personalized Cancer Treatment (CPCT-02) biopsy protocol (NCT01855477). The study protocol was approved by the medical ethics review board of the University Medical Center Utrecht, the Netherlands. Written informed consent was obtained from all participants prior to inclusion in the trial. The study population consisted of 288 patients who were scheduled for 1st or 2nd line palliative systemic treatment. Fresh-frozen metastatic tumor biopsies and matched normal blood samples were collected from 256 patients in a standardized manner[29]. WGS was successfully performed for 184 patients. Seventy patients started a new line of pembrolizumab monotherapy and were included in the current analysis. Matched RNA-seq was available for 41 patients, and immunofluorescence stainings were performed for 20 of these patients. WGS, RNA-seq and clinical data are available through the Hartwig Medical Foundation at https://www.hartwigmedicalfoundation.nl, under request number DR-176. A summary of all genomic, transcriptomic and immunofluorescence staining results as well as clinical data and response to treatment are available in Supplementary Data 2.

### Treatment and assessment of response
Patients were treated with pembrolizumab, 200 mg intravenously every 3 weeks, or 400 mg every 6 weeks. Tumor response evaluation was performed using computed tomography every 12 weeks. Treatment response was measured according to response evaluation criteria in solid tumors (RECIST) v1.1. Data cut-off was set at July 1st, 2020, resulting in a minimal follow-up of 6 months for all patients with a response to treatment. Response was assessed at 6 months of therapy and patients were classified as responder when they showed ongoing complete or partial response, or stable disease. Patients were classified as non-responder when they had progressive disease within 6 months after treatment initiation. Patients treated beyond initial radiological disease progression were classified according to the date of their first radiological progression event.

### PD-L1 immunohistochemistry and scoring
PD-L1 expression was assessed on metastatic tumor biopsies (paraffin embedded) that were freshly obtained prior to start of pembrolizumab ($n = 32$) using the companion diagnostic assay of pembrolizumab (PD-L1 IHC 22C3 pharmDx, Agilent Technologies, Carpinteria, CA, USA). When no fresh tumor biopsy was available, archival tumor tissue (primary tumor or metastasis) was used ($n = 8$). All tissues were assessed for the PD-L1 combined positivity score (CPS) by an expert genitourinary pathologist (GJLHvL).

### Whole-genome sequencing and analysis
Alignment to the human genome hg19 and pre-processing of WGS data, including the estimation of tumor purity (PURPLE v2.49), were performed using tools developed by the Hartwig Medical Foundation (https://github.com/hartwigmedical/hmftools)[29]. Subsequent detection of driver genes, mutational signatures, genomic subtypes, homologous recombination (HR) deficiency, copy number, structural variants, chromothripsis events and apolipoprotein B mRNA-editing enzyme, catalytic polypeptide-like (APOBEC) mutagenesis were estimated using scripts developed specifically to analyze WGS data from the CPCT-02 study (R2CPCT v0.4; https://github.com/J0bbie/R2CPCT)[30,36]. APOBEC-enriched tumors were classified as high when enrichment ($E$) for APOBEC-related mutations was $E \geq 3$, medium when $2 \leq E < 3$ and low when $E < 2$. The transcriptomic subtype of each sample was identified as a result of the highest ranked association between the mean (normalized) expression of all genes and a particular subtype across all subtypes[30]. The clonal fraction of mutations was estimated accounting for copy number, purity and read counts[37]. In this study, mutations were considered clonal when the variant copy number was >0.75.

### RNA-sequencing
Alignment to the human genome hg19 and GENCODE v35 (Trimmomatic v0.39 and STAR 2.7.6a), pre-processing of RNA-seq data (sambamba v0.7.0, FeatureCounts v1.6.3, RSEM v1.3.1), transcript normalization with variance stabilizing transformation, and subsequent analysis of pathway activity, and immune cell abundance were performed as described for the CPCT-02 mUC cohort[30].

### Gene signatures and the T cell-to-stroma enrichment score
A list of 36 gene signatures representing immune and stromal resident cells and their products was built from previously published resources (Supplementary Table 2 and Supplementary Data 3). Normalized gene expression levels were median centered, and the signature score was calculated as the mean expression of all genes per signature. Hierarchical clustering of gene signatures (Fig. 4) using ConsensusClusterPlus v1.54.0[38] showed that cluster one, enriched for responders, had a high signature score for immune cells and a low signature score for stromal resident cells and their products. Vice versa, cluster three with only non-responders, had a low signature score for immune cells and a high signature score for stromal resident cells and their products. Even though this result extended earlier findings that signatures of immune versus stromal resident cells and their products have differential predictive value, the contribution of individual signatures to the identified cluster of patients may vary considerably. When applying hierarchical clustering, we identified a group of signatures for T cells (Cytotoxic CD8 T cell, T cell inflamed GEP, tGE8, T cell signature, IFN gamma, Immune gene signature and chemoattractants) and stromal resident cell and products (Stromal signature, Fibroblasts, EMT/stroma core genes, CAF, TBRS) with a highly similar transcriptomic profile (Supplementary Fig. 4). In addition, these specific signatures also had high discriminatory abilities reflected in high standard deviations across samples and predictive values as shown by the AUC of ROC curves for response to pembrolizumab (Supplementary Table 3). To assess and weigh the

contribution of these two groups of signatures, the overall mean of the selected signature scores for T cells and stromal resident cells and products was calculated. These two metrics were considered to represent the global signatures for T cells and stromal resident cells and products, and at the same time filter out the noise that individual signatures may have. These two global signature scores had independent predictive power for responders (T cells: Coefficient = 3.03; $p = 0.005$) and non-responders (Stromal resident cells and products: Coefficient = −2.40; $p = 0.010$) according to multivariate logistic regression analysis. Remarkably, the arithmetic difference between these 2 global signatures (T cells minus stromal resident cells and products), which captures the concept that stromal resident cells and their products may pose a barrier to T cells, showed a significantly improved predictive value when compared to either single global signatures or individual gene signatures (Supplementary Table 3). This new metric was named the T cell-to-stroma enrichment (TSE) score because a positive TSE score points to an enrichment for T cells, while a negative TSE score points to an enrichment for stromal resident cells and their products. In fact, this metric emphasizes such enrichments as the normalized gene expression data which are raw counts transformed on the log2 scale[39]. Finally, we stratified patients into three groups according to their TSE score. The TSE score = 0.5 was selected as cut-off with which the three groups of patients obtained resembled the original clusters from Fig. 4. Patients with a TSE score ≥0.5 were considered to have a positive TSE score, patients with a TSE score ≤−0.5 were considered to have a negative TSE score and other patients were considered to have a neutral TSE score.

It is noteworthy that tumor purity does not act as a confounder for the TSE score. First, tumor purity is negatively affected by the presence of non-tumor cells, which is alike for T cells or resident stromal cells as evidenced by similar negative correlations between tumor purity and either one of the global signatures (Supplementary Fig. S13). Second, the TSE score automatically corrects for tumor purity since the former is calculated per patient and inherently represents enrichments of either T cells or resident stromal cells and their products.

### TCR repertoire

RNA-seq data was processed with MiXCR v3.0.13[40] to estimate the TCR repertoire (true) diversity and clonality. Samples with >100 total TCR reads were considered for downstream analysis. The relative proportion ($R$) was used to group clonotype sizes as follows: rare when only one read supported a clonotype; infrequent when $R < 1\%$; frequent when $R = 1$–10%; and hyper-frequent when $R > 10\%$.

### Immunofluorescence staining, imaging and analysis of T cell and stromal markers

We performed immunofluorescence stainings using whole slides of 20 patient samples of which paired RNA-seq data was available (TSE positive, $n = 7$; TSE neutral, $n = 8$; TSE negative, $n = 5$). We stained for T cells (CD4 and CD8 T cells) and stromal cells (FAP and PDPN) using markers that were considered representative for the gene signatures used to build the TSE score at RNA level. To this end, a second biopsy, which was obtained from the same lesion and collected at the same time as the first biopsy, was formalin-fixed and paraffin-embedded. Stainings for DAPI, CD3 and CD8 were obtained from multiplexed immunofluorescence performed using OPAL reagents (Akoya Biosciences, Marlborough, MA, USA) on 4 μm sections. Slides were scanned and images were obtained using VECTRA 3.0 (Akoya Biosciences), after which at least 4 stamps (regions of interest; stamp size: 671 × 500 μm²; resolution: 2 pixel/μm; pixel size: 0.5 × 0.5 μm²) were set in non-necrotic areas to cover >90% of tissue area. Images were spectrally unmixed using inForm® software (v2.4.8; Akoya Biosciences) to visualize the above markers as well as autofluorescence. Subsequently, images were manually analyzed using an in-house generated python-based image interface that had been previously tested in mUC

samples[35]. CD3+CD8+ cells were phenotyped as CD8 cells, and CD3+CD8− cells were phenotyped as CD4 cells, and their densities were calculated by dividing the number of cells by the tissue area. In two cases (one TSE positive and one TSE negative), quantification of CD4 and CD8 cells failed. In addition, consecutive sections were stained for FAP (EPR20021, Abcam)−FAM (#760-243, Ventana), PDPN (D2-40, Cell Marque)−Cy5 (Roche Applied Science), and DAPI. These slides were scanned using a Zeiss microscope (Zeiss) and the regions of interest corresponding to the above T cell markers were exported using the Qupath software (v0.4.1). Image analysis was again performed using an in-house generated python-based user interface[35]. In short, tissue areas were determined by performing gaussian blurring on the DAPI channel with a kernel size of 30 pixels, and manually thresholding this image. The thresholding for FAP and PDPN-positive areas was also performed manually using raw images corrected for background signal. Background correction for FAP images was performed via subtraction of uniform filtered images with a filter size of 500 from the original images. As the PDPN intensity was relatively uniform, background correction was not performed. Percentages of marker-positive areas were determined by dividing the areas positive for either marker by the total tissue area. Outcomes of individual T cell and stromal markers were used to generate a TSE-protein score in analogy to the TSE-RNA score. Values of the 4 protein markers were log10 transformed as $\log 10(1 + value)$, after which the stromal markers (FAP, PDPN) were subtracted from the T cell markers (CD4, CD8).

### Statistics and reproducibility

Analyses were performed using the platform R v4.1.0[41]. The Fisher's exact test was used for comparison of categorical values between groups. The Wilcoxon-rank sum test and the Kruskal−Wallis test by ranks were used for comparison of 2 or >2 groups with continuous variables, respectively. Differential expression analysis of transcripts was performed using the Wald test with DESeq2 v1.32.0[39]. A gene list of differentially expressed genes was supplied to ReactomePA v1.44.0[42] to estimate p-values by hypergeometric distribution. DeLong's and log-rank tests were used for comparing receiver operating characteristics (ROC) and Kaplan−Meier survival curves, respectively. For multivariate analyses, the Cox proportional hazards regression analysis and the logistic regression analysis were applied. $p$ values were adjusted for multiple testing using the Benjamini−Hochberg method. No statistical method was used to predetermine sample size and samples were excluded if they did not meet the inclusion criteria as depicted in Fig. 1a. This study was not randomized and the investigators were not blinded to sample annotation during outcome assessment.

### Reporting summary

Further information on research design is available in the Nature Portfolio Reporting Summary linked to this article.

## Data availability

Access to WGS and RNA-seq data were granted under request number DR-176 via the Hartwig Medical Foundation. Raw WGS and RNA-seq, and processed WGS data are freely available for academic use through standardized procedures at https://www.hartwigmedicalfoundation. nl. Clinical data with pseudonymized patient IDs are provided in the Source Data file. Processed data from the IMvigor210[20] cohort was publicly accessed without restriction at http://research-pub.gene.com/ IMvigor210CoreBiologies/. Data from the ABACUS cohort were accessed by contacting directly the corresponding author of the study[18] and are available at https://ega-archive.org/studies/ EGAS00001004445. Source Data for figures, including quantification of protein markers from tissue stainings, are provided with this paper. Source data are provided with this paper.

## Code availability

The pipeline for alignment and pre-processing of WGS data developed by the Hartwig Medical Foundation are deposited at https://github.com/hartwigmedical/hmftools. A classifier and script to calculate the TSE score from RNA normalized counts are available at https://github.com/ANakauma/TSEscore_ICIs. Additionally, the version v1.0.0 of the code used for this study is available at Zenodo (https://zenodo.org/records/10055421)[43].

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

## Acknowledgements

This publication and the underlying research are partly facilitated by Hartwig Medical Foundation and the Center for Personalized Cancer Treatment (CPCT) which have generated and made available data for this research. We thank all local principal investigators and the nurses of all contributing centers for their help with patient recruitment. We are particularly grateful to all participating patients and their families. Funding for this research was provided by The Dutch Research Council (NWO) under grant-number 848050012 (M.P.L.) and by Merck Sharp & Dohme (M.P.L.).

## Author contributions

Conceptualization: MPL, RD, and DGJR; Methodology: MR, JAN, AAMO, MPL, and RD; Data analysis: JAN, HEB, and AGJ; Data interpretation: MPL, RD, DGJR, MR, and JAN; Clinical data collection and interpretation: MR, MPL, RD, DGJR, MJBA, JLB, PH, MSvdH, BES, GJLHvL, NM, JV, HMW, RdW, and AAMvdV. Immunohistochemistry: AAMO, HEB, GJLHvL, and MR; Writing—original draft: all authors; Writing—review and editing: MR, JAN, MPL, and RD. All authors read and approved the final manuscript.

## Competing interests
M.P.L. has received research support from JnJ, Sanofi, Astellas and MSD, and consultancy fees from Incyte, Amgen, JnJ, Bayer, Servier, Roche, INCa, Pfizer, Sanofi, Astellas, AstraZeneca, Merck Sharp & Dohme, Novartis, Julius Clinical and the Hartwig Medical Foundation (all paid to the Erasmus MC Cancer Institute). D.G.J.R. has received research support from Treatmeds and consultancy fees from Bristol-Myers Squibb, Bayer, AstraZeneca, Merck, Pfizer (all paid to the Erasmus MC Cancer Institute). R.d.W. has received consultancy fees from Sanofi, Merck, Astellas, Bayer, Hengrui and Orion, speaker fees from Sanofi and Astellas, research support from Sanofi and Bayer (all paid to the Erasmus MC Cancer Institute). A.A.M.v.d.V. has received consultancy fees from for BMS, MSD, Merck, Novartis, Roche, Sanofi, Pierre Fabre, Ipsen, Eisai, Pfizer (all paid to the Erasmus MC Cancer Institute). M.S.v.d.H. has received research support from Bristol-Myers Squibb, AstraZeneca and Roche, and consultancy fees from Bristol-Myers Squibb, Merck Sharp & Dohme, Roche, AstraZeneca, Seattle Genetics and Janssen (all paid to the Netherlands Cancer Institute). J.L.B. has received research support from Decipher Biosciences and Merck Sharp & Dohme, and consultancy fees from Merck Sharp & Dohme, Eight Medical, Ambu, APIM therapeutics, Bristol-Myers Squibb, Astellas Roche and Janssen (all paid to the Erasmus MC Cancer Institute). N.M. has received research support from Astellas, Janssen, Pfizer, Roche and Sanofi Genzyme, and consultancy fees from Roche, MSD, BMS, Bayer, Astellas and Janssen (all paid to the Radboud University Medical Center). H.M.W. has received consultancy fees from Roche and Astellas (all paid to the Amphia hospital, Breda). P.H. has received consultancy fees from Astellas, Merck Sharp & Dohme, Pfizer AstraZeneca, Bristol-Myers Squibb and Ipsen (all paid to the Franciscus Gasthuis & Vlietland Hospital, Rotterdam/Schiedam). M.J.B.A. has received advisory board/consultancy honoraria from Amgen, Bristol Myers Squibb, Novartis, MSD-Merck, Merck-Pfizer, Pierre Fabre, Sanofi, Astellas, Bayer, research grants from Merck-Pfizer (all paid to Maastricht UMC+ Comprehensive Cancer Center). G.J.L.H.v.L. has received research grants from Roche and AstraZenaca, and has been member of advisory boards of Roche and Merck (all paid to the Erasmus MC Cancer Institute). R.D. has received research support from MSD and Bayer, personal fees from Bluebird Bio, Genticel, other support from Pan Cancer T outside the submitted work (all paid to the Erasmus MC Cancer Institute), and is listed as inventor for European patent application nos. 21152822.9 and 21184727.2 (pending to Erasmus MC). M.R., J.A.N.-G., A.G.-J., J.V., A.A.M.O. and H.E.B. declare no competing interests.

## Additional information

[1]Department of Medical Oncology, Erasmus MC Cancer Institute, University Medical Center Rotterdam, Rotterdam, The Netherlands. [2]Department of Urology, Erasmus MC Cancer Institute, University Medical Center Rotterdam, Rotterdam, The Netherlands. [3]Cancer Computational Biology Center, Erasmus MC Cancer Institute, University Medical Center Rotterdam, Rotterdam, The Netherlands. [4]Department of Molecular Carcinogenesis, The Netherlands Cancer Institute, Amsterdam, The Netherlands. [5]Oncode Institute, Utrecht, The Netherlands. [6]Department of Medical Oncology, GROW—School for Oncology and Reproduction, Maastricht University Medical Center, Maastricht, The Netherlands. [7]Department of Medical Oncology, Franciscus Gasthuis & Vlietland Hospital, Rotterdam/Schiedam, The Netherlands. [8]Department of Medical Oncology, The Netherlands Cancer Institute, Amsterdam, The Netherlands. [9]Barts Cancer Institute, Queen Mary University of London, London, UK. [10]Department of Pathology, Erasmus MC Cancer Institute, University Medical Center Rotterdam, Rotterdam, The Netherlands. [11]Department of Medical Oncology, Radboud University Medical Center, Nijmegen, The Netherlands. [12]Department of Medical Oncology, Amsterdam UMC, Vrije Universiteit Amsterdam, Cancer Center Amsterdam, Amsterdam, The Netherlands. [13]Department of Internal Medicine, Amphia Hospital Breda, Breda, The Netherlands. [14]Department of Radiology & Nuclear Medicine, Erasmus MC Cancer Institute, University Medical Center Rotterdam, Rotterdam, The Netherlands. [15]Present address: Amgen Inc., Breda, The Netherlands. [16]These authors contributed equally: Maud Rijnders, J. Alberto Nakauma-González. [17]These authors jointly supervised this work: Reno Debets, Martijn P. Lolkema. ✉e-mail: j.debets@erasmusmc.nl

