## [Peer Review File · Nature Communications]

Gene-expression-based T-Cell-to-Stroma Enrichment (TSE) score predicts response to immune checkpoint inhibitors in urothelial cancerREVIEWER COMMENTS

Reviewer #1 (Remarks to the Author):

Rijnders et al. present an interesting approach to identify biomarkers for better stratifying patients with regard to therapeutic response to ICIs in metastatic bladder cancer. Although this addresses a great need in the field, there are certain limitations in this study that weaken the overall conclusions. The most important limitation is the small cohort size and the lack of histological validation of the findings. This becomes an even larger issue in light of the fact that sampling is based on needle biopsies which results to a great probability of molecularly characterizing nonrepresentative tumor samples. It is puzzling that the authors choose not to verify some of their findings using paraffin sections which are definitely available as the pD-L1 expression analysis is performed on paraffin sections.

Specific comments:

1. Based on Supplementary Table 1, the number of responders is quite low. Among those, the PD-L1 combined positivity score is positive in 5, negative in 8 and unknown in 11. Overall, the unknown combined positivity score is unknown in 46% of responders and 41% of non-responders. In other words, the CPS is calculated on the basis of 30 patients (11 and 19, respectively). This is too small a cohort, to conclude that CPS (with its known limitations) is not a good biomarker in this cohort.

2. In Supplementary Figure 3, the authors do not describe the genes included in each of the categories, i.e., which genes were included in the "cell cycle" process, the "RTK" and "DDR" groups. Moreover, although the results might not be statistically significant, this is probably due to the small sample size. For example, the representation of tumors with mutations in the RTK-RAS, WNT, PI3K groups is substantially higher in the responders. What are the statistical values?

3. What is the overlap between the 41 cases with RNA-seq data and the 40 cases with a PD-L1 CPS score?

4. In Figure 4, the 3 groups seem a little arbitrary and not homogeneous. For example, in the first group, there are cases that look much more similar with cases in the other groups. I understand that this is computational, but could it be that some of the categories used should not be part of the analysis? The authors should comment on this.

5. Also in Figure 3, the majority of biopsies in the first cluster comes from LNs (13/18), while the LN biopsies in the other groups are the minority (1/10 and 4/13, for clusters 2 and 3, respectively). Is there bias for immune cell enrichment here? In fact, the responders seem to overlap a lot with the LN biopsies and to be inversely correlated with the tumor purity. There seems to exist a bias here. Could it be that the better survival correlates with locally advanced disease while patients with distant metastases do not benefit? The authors mention the fact that there is a positive correlation with LN biopsies but do not further comment on this. Is there any normalization for tumor content? If not, immune cell over-representation might be misleading.

6. In the large IMvigor210 cohort, the TMB AUC is superior (0.71) to the TSE score (0.65), albeit non-significant. In this cohort, ICIs were given as a second line therapy. However, this is a much larger cohort than the one described in this manuscript, and although the usefulness of the TSE is not challenged, it does not seem to be superior to the TMB, further supporting the possibility that the small cohort size and the heterogeneous origin of the biopsies, might bias the analysis and its findings. The small size is a limitation that authors acknowledge as well.

7. Another limitation of this study is that the high infiltration of T cells (and dendritic cells) in the high TSE and the high content of stromal cells (in the low TSE score) are not verified histologically. Although the authors measure tumor purity, it is not clear how this is calculated. In any case, immunostainings are missing in this study.

Reviewer #2 (Remarks to the Author):

In this manuscript, the statistical methods are adaptive. However, I have the following questions:

1. please give the comparison analysis of patient characteristics between responder and non-responder in Supplementary Table 1 (i.e., p-values).

2. Are the patient overall and progression-free survival times relative to patient characteristics in Supplementary Table 1?

3. Does the TSE score depends on patient characteristics?

4. Section " Statistical Analysis" could be revised as:

"Fisher's exact and Wilcoxon-rank sum tests were used for comparison between groups and the t-test was used for comparisons with continuous variables. DeLong's and log-rank tests were used for comparing receiver operating characteristics (ROC) and Kaplan-Meier survival curves, respectively. For multivariate analyses, the Cox proportional hazards regression analysis and the logistic regression analysis were applied".

Reviewer #3 (Remarks to the Author):

This manuscript by Rijnders et al aims to identify effective biomarkers of response to immune checkpoint inhibitors (ICIs) which is a major undertaking and to be applauded, since this is an urgent unmet clinical need in the field. Using gene expression data from RNAseq, they defined a T cell-to-stroma enrichment (TSE) score—a signature-based metric as a surrogate for the abundance of T cells to stromal cells. Remarkably, they demonstrated that patients with positive TSE score showed 67% PFS at 6 months and those with negative TSE score showed 0% PFS. This is further validated by two independent cohorts IMvigor210 (n=348) and ABACUS (n=84).

However, there are certain technical weakness that might introduce bias.

1. For instance, the original data were generated primarily from metastatic sites (e.g., lymph node LN n=31, lung n=3, liver n=14 etc), instead of the primary tumor (only n=5) hampering the overall data interpretation, since it is generally expected that the T cell and stromal content are very different in these different organ sites compared to those in the primary tumor. Importantly, tissues collected from LN represent a major group of responders, it is therefore not surprising that a high T-to-stroma ratio exist in the LN, compared with liver predominant (in the nonresponder group)

- It is problematic to draw a strong conclusion unless they restrict on LN samples only or liver only and perform similar analysis comparing TSE score to reach the same conclusion regarding TSE score comparing responder vs nonresponders within the same organ (we understand n number is a potential issue)

- It might be scientifically relevant to highlight TCR clonal diversity in their current study, since TCR clonality would not necessarily be impacted by specific organ sites—which is currently not a major highlight (and the link to dendritic cells is indeed more interesting, implicating priming of antigen specific T cells). Would this be a point that can be further developed?

2. 90% of patients received pembro as second-line therapy, but a significantly more responders are found in patients receiving pembro as first-line (25%) vs second-line 2%, suggesting pretreatment with other therapy e.g., chemo and associated TME change could be a driver for ICI response (which was not taken into account in the current study). Do they have sequential tissues to address such phenomenon? Or can such analysis be further studied using the IMvigor210 cohort?

3. Upregulated genes in responders include IFN γ signaling, trafficking and processing of endosomal TLR, which are quite interesting and relevant. However IL-10 signaling was also amongst the upregulated gene category, which is slightly unexpected. Are these negative regulators toward IL-10 signaling, since IL-10 is known to be a major immunosuppressive pathway? Might be important to discuss further.

4. It has been reported previously that genes related to ECM organization and collagen formation (i.e., Figure 3 vs Nat Commun. 2018 Aug 29;9(1):3503 and Nat Rev Urol. 2022 Sep;19(9):515-533) are associated with ICI response, which is encouraging but not necessarily innovative. Can they delineate further what are the similarity and difference to the 2018 and 2022 studies in these gene categories to highlight the novelty?

5. It is a general issue when analyzing bulk RNAseq data that samples with high T cell scores are often also high in another immunosuppressive cell category (e.g., Figure 4) when digital deconvolution is not applied. Is there room for more sophisticated bioinformatic analysis in Figure 4?

6. Minor comment: Figure 6 is slightly heterogeneous to read, e.g., some cohorts showing OS but not RFS vice versa. It might be important to show both data although certain survival data might be significant/not statistically significant in certain cohorts.

Response to reviewers' comments

Reviewer #1

Rijnders et al. present an interesting approach to identify biomarkers for better stratifying patients with regard to therapeutic response to ICIs in metastatic bladder cancer. Although this addresses a great need in the field, there are certain limitations in this study that weaken the overall conclusions. The most important limitation is the small cohort size and the lack of histological validation of the findings. This becomes an even larger issue in light of the fact that sampling is based on needle biopsies which results to a great probability of molecularly characterizing nonrepresentative tumor samples. It is puzzling that the authors choose not to verify some of their findings using paraffin sections which are definitely available as the PD-L1 expression analysis is performed on paraffin sections.

Response

*We thank R#1 for assessing our study and for the questions asked. We fully concur that sample size and needle-based sampling could introduce bias to our analysis. Increasing the sample size of our cohort could potentially mitigate such bias. The inclusion of patients in this study stopped on 31st December 2020, which unfortunately makes this option not feasible. An alternative solution to discard any bias introduced by our patient cohort, as proposed by the authors, is the inclusion of independent cohorts to validate the TSE score as a predictive marker of response to ICI. To this end, we have introduced 2 independent patient cohorts, namely the IMVigor210 and ABACUS cohorts, and again demonstrated the predictive value of the TSE score (**see Result section and Figure 7**). Any bias (sampling, sample size or patient characteristics) that we may have introduced in our discovery cohort has passed the test in these two independent validation cohorts. Specifically regarding needle-based sampling, these biopsies were used to assess tumor characteristics. Prior to whole genome sequencing analysis and tissue stainings, these samples were used to estimate the tumor cell percentage and/or stain for H&E stainings to confirm that only representative samples were used for further analyses.*

*To address R#1's concern regarding the lack of histological validation, we have performed immunofluorescence stainings using whole slides of 20 patient samples of which paired RNA-seq data was available (TSE positive, n=7; TSE neutral, n=8; TSE negative, n=5). We have stained for T cells (CD4 as well as CD8 T cells) and stromal cells (Fibroblast activating protein (FAP) as well as Podoplanin (PDPN)) using markers that were considered representative for the gene signatures used to generate the TSE score at the RNA level. Visualization and quantification of the densities of these markers extended our understanding of the TSE score, and enabled the generation of a TSE-protein score in analogy to the TSE-RNA score; **see revised Materials and Methods and new Supplementary Figure 11** for details. We observed a high T cell density in tumor areas with low stromal density, and vice versa, we observed low T cell density in tumor areas with high stromal density (**see new Figure 6a**). Furthermore, the TSE-protein score correlated with its parental RNA counterpart, and was able to predict response to pembrolizumab (**see new Figure 6b**). New findings have been described and discussed in the revised manuscript in lines 226-238 and 327-332 respectively.*

Specific comments:

1. Based on Supplementary Table 1, the number of responders is quite low. Among those, the PD-L1 combined positivity score is positive in 5, negative in 8 and unknown in 11. Overall, the combined positivity score is unknown in 46% of responders and 41% of non-responders. In other words, the CPS is calculated on the basis of 30 patients (11 and 19, respectively). This is too small a cohort, to conclude that CPS (with its known limitations) is not a good biomarker in this cohort.

Response

We thank R#1 for this comment and we would like to clarify our conclusion. The number of samples with a known CPS score is 40, consisting of 16 positive and 24 negative scores. These group sizes are comparable to the group sizes we identified for the different TSE categories (positive, n=15; neutral, n=14; and negative, n=12). For each measure (CPS or TSE), the proportions of responders among the different scores (2 for CPS, and 3 for TSE) were tested for statistically significant difference. Though the number of samples is low, we reached statistical significance with the TSE score, but not with the CPS score. We agree that a larger cohort would have been preferable and may have shown predictive value for CPS, and have included this interpretation in lines 287-289 of the revised discussion section.

2. In Supplementary Figure 3, the authors do not describe the genes included in each of the categories, i.e., which genes were included in the "cell cycle" process, the "RTK" and "DDR" groups. Moreover, although the results might not be statistically significant, this is probably due to the small sample size. For example, the representation of tumors with mutations in the RTK-RAS, WNT, PI3K groups is substantially higher in the responders. What are the statistical values?

Response

*The list of all genes, as well as their categorization by group/pathway, used in this analysis is now available in **Supplementary Data 1**. Also, p-values are now indicated in the figure.*

3. What is the overlap between the 41 cases with RNA-seq data and the 40 cases with a PD-L1 CPS score?

Response

*In 21 cases, PD-L1 CPS score and RNA-seq were available for the same tumor. This is not explicitly mentioned, but it can be inferred from **Figure 4** (also available in Source Data for Figure 4).*

4. In Figure 4, the 3 groups seem a little arbitrary and not homogeneous. For example, in the first group, there are cases that look much more similar with cases in the other groups. I understand that this is computational, but could it be that some of the categories used should not be part of the analysis? The authors should comment on this.

Response

Figure 4 is the result of hierarchical clustering that was applied to group patients in an unbiased manner. This method works well for most samples, but indeed as justly noticed by

R#1, some cases are more challenging to cluster. The groups of patients obtained from this unbiased analysis indicated that the signatures that encompass T cells and those that encompass stromal resident cells and products directed the clusters, and to a lesser extent the signatures that encompass immune cells (other than T cells). From this observation, we defined the TSE score and were able to assign each patient sample to each of 3 possible TSE categories (i.e., positive, neutral or negative). Although there is a high concordance between the clusters and TSE categories, some patients were assigned to a TSE category that did not match the cluster, which is in line with the observation of the reviewer. The TSE categories, and not the clusters, were finally used to further assess and validate the predictive value for response to ICI.

5. Also in Figure 4, the majority of biopsies in the first cluster comes from LNs (13/18), while the LN biopsies in the other groups are the minority (1/10 and 4/13, for clusters 2 and 3, respectively). Is there bias for immune cell enrichment here? In fact, the responders seem to overlap a lot with the LN biopsies and to be inversely correlated with the tumor purity. There seems to exist a bias here. Could it be that the better survival correlates with locally advanced disease while patients with distant metastases do not benefit? The authors mention the fact that there is a positive correlation with LN biopsies but do not further comment on this. Is there any normalization for tumor content? If not, immune cell over-representation might be misleading.

Response

*We concur with R#1 that immune cell enrichment for samples from LNs, and a potential bias towards TSE score, needs further analysis and explanation. When only samples from LN or liver are considered (n=18 and 9, respectively), we show that the TSE score is able to predict response to pembrolizumab independent of metastatic site (**new Supplementary Figure 5**). Also, see response to R#3 Comment 1 for more details. We cannot exclude that LN sites yield positive TSE scores more frequently, and have commented on this in the revised discussion section (lines 277-278), yet in the authors' view this does not introduce a site-selective bias to the predictive value of the TSE score towards the patient's response to ICI.*

*Regarding tumor purity, it is remarkable that signatures for both T cells and stromal resident cells and products negatively correlate with purity (**new Supplementary Figure S13**), suggesting that for all patient samples the relative contribution of non-tumor cells is similar. In other words, authors argue that both categories of signatures are affected equally (by a constant factor) by tumor purity of each sample, excluding the need to further correct for this parameter. We have added a short explanation of this in lines 468-474.*

6. In the large IMvigor210 cohort, the TMB AUC is superior (0.71) to the TSE score (0.65), albeit non-significant. In this cohort, ICIs were given as a second line therapy. However, this is a much larger cohort than the one described in this manuscript, and although the usefulness of the TSE is not challenged, it does not seem to be superior to the TMB, further supporting the possibility that the small cohort size and the heterogeneous origin of the biopsies, might bias the analysis and its findings. The small size is a limitation that authors acknowledge as well.

Response

Indeed, the IMvigor210 cohort is a great data set to validate our findings, particularly regarding treatment (atezolizumab) in 348 patients with platinum-refractory locally advanced or mUC. Nevertheless, the IMvigor210 cohort also has limitations that could impact interpretation. First, 50% of samples were collected one year or more prior to treatment initiation. Contrary to genomic mutations, the transcriptome is more dynamic and prone to change (for instance upon other treatments or tumor progression). Second, 83% of samples were from the urinary tract, probably from the primary tumor, and it is currently unclear how similar or non-similar the transcriptomic profiles of the primary tumor versus metastatic sites are. Third, response to treatment was defined as the best overall response, which differs from the definition we used in our cohort, which was response at 6 months after start treatment. All of these variables, besides differences in drug and disease stages of the patient cohorts, may have contributed to the discrepancies observed between the IMvigor210 and our cohort. Notably, these differences primarily related to the predictive value of TMB, whereas the predictive value of the TSE score was robust among the discovery cohort and the 2 validation cohorts. We briefly summarized above points in lines 343-346.

7. Another limitation of this study is that the high infiltration of T cells (and dendritic cells) in the high TSE and the high content of stromal cells (in the low TSE score) are not verified histologically. Although the authors measure tumor purity, it is not clear how this is calculated. In any case, immunostainings are missing in this study.

Response

Authors have addressed the lack of histological validation, and have performed immunofluorescence stainings using whole slides of 20 patient samples of which paired RNA-seq data was available. In short, visualization and quantification of protein markers enabled the generation of a TSE-protein score; **see revised Materials and Methods and new Supplementary Figure 11**. We observed high T cell density in tumor areas with low stromal density (TSE positive samples), and vice versa, we observed low T cell density in tumor areas with high stromal density (TSE negative samples; **see new Figure 6a**). See above response to main comment for more details.

The tumor purity can be considered as an indirect measure of the presence of non-tumor cell types. The tumor purity was automatically generated by the pipeline of the Hartwig Medical Foundation (<https://github.com/hartwigmedical/hmftools>). In short, the purity ploidy estimator (PURPLE v2.49) algorithm combines the B-allele frequency (BAF), depth ratios and genomic alterations to estimate the purity and copy number of tumor samples from whole-genome sequencing data. The assessment of tumor purity, including the corresponding reference, has been added in the revised materials and Methods, lines 407-408.

Reviewer #2

In this manuscript, the statistical methods are adaptive. However, I have the following questions:

1. Please give the comparison analysis of patient characteristics between responder and non-responder in Supplementary Table 1 (i.e., p-values).

Response

We thank R#2 for the assessment of our manuscript. Along R#2's recommendation, we have updated **Supplementary Table 1** with p-values for all relevant comparisons between responders and non-responders.

2. Are the patient overall and progression-free survival times relative to patient characteristics in Supplementary Table 1?

Response

We have now added p-values for the associations between OS and PFS and clinical characteristics included in **Supplementary Table 1**.

3. Does the TSE score depend on patient characteristics?

Response

We have tested the dependency of the TSE score on patient characteristics, and did see a correlation between neutral TSE score and samples from female patients (Fisher's exact test $p = 0.004$). Other characteristics such as age ($p=0.64$, Kruskal-Wallis test) and pre-treatment ($p=0.54$, Fisher's exact test) did not correlate with TSE categories. We now mention these findings in lines 196-199. All patient characteristics, response to treatment and TSE categories are listed in **Supplementary Data 2**, including results of inter-dependency tests.

4. Section " Statistical Analysis" could be revised as:

"Fisher's exact and Wilcoxon-rank sum tests were used for comparison between groups and the t-test was used for comparisons between continuous variables. DeLong's and log-rank tests were used for comparing receiver operating characteristics (ROC) and Kaplan-Meier survival curves, respectively. For multivariate analyses, the Cox proportional hazards regression analysis and the logistic regression analysis were applied".

Response

We thank R#2 for this valuable suggestion. We have adapted the text accordingly.

Reviewer #3

This manuscript by Rijnders et al aims to identify effective biomarkers of response to immune checkpoint inhibitors (ICIs) which is a major undertaking and to be applauded, since this is an urgent unmet clinical need in the field. Using gene expression data from RNAseq, they defined a T cell-to-stroma enrichment (TSE) score—a signature-based metric as a surrogate for the abundance of T cells to stromal cells. Remarkably, they demonstrated that patients with positive TSE score showed 67% PFS at 6 months and those with negative TSE score showed 0% PFS. This is further validated by two independent cohorts IMvigor210 (n=348) and ABACUS (n=84).

However, there are certain technical weakness that might introduce bias.

1. For instance, the original data were generated primarily from metastatic sites (e.g., lymph node LN n=31, lung n=3, liver n=14 etc), instead of the primary tumor (only n=5) hampering the overall data interpretation, since it is generally expected that the T cell and stromal content are very different in these different organ sites compared to those in the primary tumor. Importantly, tissues collected from LN represent a major group of responders, it is therefore not surprising that a high T-to-stroma ratio exist in the LN, compared with liver predominantly (in the non-responder group).

It is problematic to draw a strong conclusion unless they restrict on LN samples only or liver only and perform similar analysis comparing TSE score to reach the same conclusion regarding TSE score comparing responder vs non-responders within the same organ (we understand n number is a potential issue).

Response

*We thank R#3 for having assessed our manuscript, and for pointing out potential technical weaknesses. Authors agree that the location of the metastatic site may yield different TSE scores, particularly for LN sites, where compared to other sites there is an enrichment for immune cells. Following R#3's suggestion, and to test whether the predictive value of the TSE score depends on metastatic site, we tested only samples from LN or liver (n=18 and 9, respectively). We observed that the TSE score is able to predict response to pembrolizumab independent of metastatic site (**new Supplementary Figure 5**). In fact, in the case of LN metastases we reached similar statistics as for the whole cohort. In the case for liver metastasis, there was a clear numerical difference between the TSE categories, with the highest and lowest proportions of responders found for the positive and negative TSE categories, yet a statistically significant difference was not reached due to the limited number of patients. Additionally, we reached the same conclusion using samples from the ABACUS cohort, which only consisted of primary tumors. Together, these results put forward that the TSE score is a robust metric that is able to predict response to ICIs independently of the metastatic site. We describe and discuss the above results in the text lines 191-194.*

2. It might be scientifically relevant to highlight TCR clonal diversity in their current study, since TCR clonality would not necessarily be impacted by specific organ sites—which is currently not a major highlight (and the link to dendritic cells is indeed more interesting, implicating priming of antigen specific T cells). Would this be a point that can be further developed?

Response

We thank R#3 for scientifically highlighting TCR diversity in relation to the predictive value of the TSE score. To better position the TCR repertoire in our revised manuscript, we have now added both TCR diversity as well as clonality to Figure 4. In line with R#3's suggestion, a positive TSE score indeed associated with a more diverse TCR repertoire and a higher number of TCR clones when compared to a negative TSE score. Zooming in to the relative contribution of clones with variable sizes, we noted that samples with a positive TSE harbored a higher fraction of infrequent clones, whereas samples with a negative TSE score harbored a higher fraction of hyper-frequent clones (**Supplementary Figures 8 and 10**). These data support the notion that in tumor tissues where there is a higher abundance of T cells over stromal resident cells, and consequently more contact areas between T cells and tumor cells, (some) expansion has occurred of a high number of T cells that cover a high breadth of TCRs, which then yields a relative dominance of infrequent TCR clones. Yet, in tumor tissues where there is a higher abundance of stromal resident cells over T cells, (some) expansion has only occurred of a restricted number of T cells, which, given the low total size of TCR clones, yields a relative dominance of hyper-frequent TCR clones.

In addition to the TCR repertoire, we have also zoomed into the relative contribution of different immune cell (non-T cell) populations, and observed that samples with a positive TSE score contained a higher fraction of dendritic cells and B cells when compared to samples with a negative TSE score (**Supplementary Figure 9**). This finding extends a previous study where we reported the clustering of such antigen-presenting cells together with T cells in metastasized lesions of patients who responded to pembrolizumab but not in lesions from non-responders (Rijnders M, Clin Can Res, 2022).

Taken together, tumors with a positive TSE score display a relatively high number of many different TCR clones, which, together with the presence of antigen presenting cells, makes these tumors prone to respond to ICI. In contrast, tumors with a negative TSE score display a relatively low number of restricted TCR clones, as well as near absence of antigen-presenting cells, particularly dendritic cells, which together compromises these tumors to respond to a multitude of antigens and makes them resistant to ICI. The above description and interpretation of distribution and number of TCR clones, and immune cell landscape, is now part of the revised discussion in lines 311-321.

3. 90% of patients received pembro as second-line therapy, but a significantly more responders are found in patients receiving pembro as first-line (25%) vs second-line 2%, suggesting pretreatment with other therapy e.g., chemo and associated TME change could be a driver for ICI response (which was not taken into account in the current study). Do they have sequential tissues to address such phenomenon? Or can such analysis be further studied using the IMvigor210 cohort?

Response

R#3 raises an interesting point regarding a potential adverse effect of pre-treatment on clinical response, which is suggestive of a change in TSE score. From our cohort, paired samples prior and after pre-treatment have not been collected, neither from the IMvigor210 cohort,

preventing an intra-patient analysis of the effect of pre-treatment towards TSE score. Nevertheless, using the IMvigora210 cohort we were able to assess whether response of pre-treated versus treatment-naïve patients changed as a function of the TSE category (this analysis was not feasible for the discovery cohort due to the limited number of treatment naïve patients (n=7)). Patients in this cohort were considered treatment-naïve when neither intravesical BCG nor platinum-based chemotherapy has previously been administered. Patients were considered pre-treated when platinum-based chemotherapy has been administered, not preceded by BCG. We observed that among non-responders there was an enrichment of TSE negative tumors in pre-treated (30/59) versus treatment-naïve patients (9/37) (Fisher's exact test $p = 0.01$; **see new Supplementary Figure 12**). This outcome suggests that platinum-based chemotherapy may induce a micro-environmental shift towards less T cells and/or more stromal resident cells, and consequently adversely impacts response to ICI as a second-line therapy. We have described and discussed the above in lines 252-257 and 336-342.

4. Upregulated genes in responders include IFN γ signaling, trafficking and processing of endosomal TLR, which are quite interesting and relevant. However, IL-10 signaling was also amongst the upregulated gene category, which is slightly unexpected. Are these negative regulators toward IL-10 signaling, since IL-10 is known to be a major immunosuppressive pathway? Might be important to discuss further.

Response

We agree that the up-regulated expression of IL10 in responders (according to **Figure 3**) deserves further explanation. Searching in all up-regulated genes in responding patients, we did not find any known regulators of the IL-10 pathway. Indeed, as pointed out by R#3, IL-10 is a recognized immunosuppressor, however, recent studies have also associated IL-10 with T cell activation in solid tumors (Saraiva et al, J Exp Med 2020; PMID: PMC7037253). Since IL-10 can be expressed by several cell types, including cancer cells, the origin as well as exact functioning of IL-10 in the context of ICI treatment requires further investigation using approaches other than bulk RNA-seq. We have added this to the result section of the revised manuscript in lines 149-153.

4. It has been reported previously that genes related to ECM organization and collagen formation (i.e., Figure 3 vs Nat Commun. 2018 Aug 29;9(1):3503 and Nat Rev Urol. 2022 Sep;19(9):515-533) are associated with ICI response, which is encouraging but not necessarily innovative. Can they delineate further what are the similarity and difference to the 2018 and 2022 studies in these gene categories to highlight the novelty?

Response

R#3 is correct in stating that T cells and stroma (including ECM and collagen) have been previously identified as predictors of ICI response, which we have acknowledged in lines 160-161. In our study, instead of defining a new signature to predict ICI response, we decided to build on and exploit gene signatures of T cells, other immune cells and stromal resident cells and products that others have reported and select the most informative ones towards predicting response. The cluster analysis based on these signatures, not on the individual genes, revealed that all these signatures highly correlate with each other and that all are able to predict ICI response (Supplementary Table 3); thereby providing further validation in our cohort and

extending the outcomes of previous reports. Next, we have captured relevant signatures into a single and novel metric, the TSE score, which combines the often inversely related T cell and stromal cell signatures, and provides a significantly improved predictive value when compared to existing single signatures. One main advantage of the TSE score over single signatures, such as the ones referred to by R#3, in metastatic disease is that its predictive value is not affected by the specific characteristics of an organ, such as the immune cell-rich lymph node (exemplified in **new Supplementary Figure 5**). Notwithstanding the fact that our findings are conceptually not new, we argue that the identification and validation of a robust predictive marker that optimally integrates existing signatures is innovative. We have added text in lines 274-282 to highlight the novelty and potential impact of our study.

5. It is a general issue when analyzing bulk RNAseq data that samples with high T cell scores are often also high in another immunosuppressive cell category (e.g., Figure 4) when digital deconvolution is not applied. Is there room for more sophisticated bioinformatic analysis in Figure 4?

Response

Authors agree that bulk RNAseq data analysis comes with challenges. To this end, we have applied immune cell deconvolution and found that the distribution of cell types was variable across the three TSE categories (**Supplementary Fig. 8-9**). As pointed out by R#3, we indeed have noted that the frequency of regulatory T cells correlated with a positive TSE score, indicative of adaptive feedback following an initial T cell response. To further investigate the exact mechanisms that underly the presence of immunosuppressive cells, we would require high-resolution data achieved by single-cell technologies, which was despite its value considered out-of-scope given the message of the current study, i.e., the presentation of a novel, robust metric to predict response of mUC patients to ICI.

6. Minor comment: Figure 6 is slightly heterogeneous to read, e.g., some cohorts showing OS but not RFS vice versa. It might be important to show both data although certain survival data might be significant/not statistically significant in certain cohorts.

Response

That is correct. Ideally, we would indeed like to show OS for both cohorts, however, OS for the ABACUS trial was not made available for this analysis. Also, the PFS was only available for 40 patients and not for all 84 patients.

REVIEWERS' COMMENTS

Reviewer #1 (Remarks to the Author):

I would like to thank the reviewers for their efforts to address all of my concerns. To the extent possible, they have achieved this and have produced a much improved manuscript. Nevertheless, the main concerns relating to cohort size and the potential bias in the tissue of the samples analyzed remain. In this reviewer's view, these are extremely important issues, particularly in the field of diagnostics.

Reviewer #2 (Remarks to the Author):

Thanks for addressing all of my comments. I have no further comments.

Response to reviewers' comments

Reviewer #1

I would like to thank the reviewers for their efforts to address all of my concerns. To the extent possible, they have achieved this and have produced a much improved manuscript. Nevertheless, the main concerns relating to cohort size and the potential bias in the tissue of the samples analyzed remain. In this reviewer's view, these are extremely important issues, particularly in the field of diagnostics.

Response

We thank again R#1 for having assessed our revised study. We acknowledge the concerns of the reviewer and for this reason, we have included a paragraph to indicate the limitations of this study in the discussion section lines 356-360.

Reviewer #2:

Thanks for addressing all of my comments. I have no further comments.

Response

We thank R#2 again for the valuable comments towards the original manuscript, which have improved this manuscript.